# Caregiver-Attributed Etiologies of Children’s Attention-Deficit/Hyperactivity Disorder: A Study in Taiwan

**DOI:** 10.3390/ijerph17051652

**Published:** 2020-03-04

**Authors:** Wen-Jiun Chou, Tai-Ling Liu, Ray C. Hsiao, Yu-Min Chen, Chih-Cheng Chang, Cheng-Fang Yen

**Affiliations:** 1College of Medicine, Chang Gung University, Taoyuan 33302, Taiwan; wjchouoe2@gmail.com; 2Department of Child and Adolescent Psychiatry, Chang Gung Memorial Hospital, Kaohsiung Medical Center, Kaohsiung 83301, Taiwan; 3Department of Psychiatry, Kaohsiung Medical University Hospital, Kaohsiung 80708, Taiwan; dai32155@gmail.com (T.-L.L.); bluepooh79@msn.com (Y.-M.C.); 4Department of Psychiatry, School of Medicine, and Graduate Institute of Medicine, Kaohsiung Medical University, Kaohsiung 80708, Taiwan; 5Department of Psychiatry and Behavioral Sciences, University of Washington School of Medicine, Seattle, WA 98195-6560, USA; rhsiao@u.washington.edu; 6Department of Psychiatry, Children’s Hospital and Regional Medical Center, Seattle, WA 98105, USA; 7Department of Psychiatry, Chi Mei Medical Center, Tainan 70246, Taiwan; 8Department of Health Psychology, College of Health Sciences, Chang Jung Christian University, Tainan 71101, Taiwan

**Keywords:** attention-deficit/hyperactivity disorder, caregivers, etiology, affiliate stigma

## Abstract

The aim of this survey study was to examine the etiologies of attention-deficit/hyperactivity disorder (ADHD) attributed by caregivers of Taiwanese children with ADHD, particularly factors affecting such attribution. This study had 400 caregivers of children with ADHD as participants. We examined the caregiver-attributed etiologies of ADHD and factors affecting such attribution. Caregivers completed the self-report questionnaire to rate how likely they perceived various etiologies of ADHD to be; the Affiliate Stigma Scale for the level of affiliate stigma; and the short Chinese version of the Swanson, Nolan, and Pelham, Version IV Scale for child’s ADHD and oppositional symptoms. Brain dysfunction (84.8%) was the most commonly attributed etiology, followed by failure of caregivers in disciplining the child (44.0%); a poor diet, such as a sugar-rich diet (40.8%); a poor living environment (38.8%); the child imitating their peers’ improper behavior (37.3%); failure of school staff in disciplining the child (29.0%); the education system’s overemphasis on academic performance (27.3%); and supernatural beings or divination-based reasons (3.8%). Caregivers’ affiliate stigma was significantly associated with the attribution of several nonbiological etiologies other than brain dysfunction. Caregivers’ education level and children’s sex, hyperactivity/impulsivity, and oppositional symptoms were significantly associated with various caregiver-attributed etiologies. Therefore, to deliver more accurate knowledge about ADHD in educational programs, health professionals should consider those etiologies that are attributed by caregivers of children with ADHD.

## 1. Introduction

### 1.1. Caregiver-Attributed Etiologies of Their Child’s ADHD

Attention-deficit/hyperactivity disorder (ADHD) is the most prevalent childhood-onset neurodevelopmental disorder. ADHD is highly heritable and multifactorial; multiple genes and non-inherited factors contribute to the disorder [1]. Neuropsychological dysfunctions such as executive dysfunction have been proposed to explain the deficits in behavioral inhibition, working memory, regulation of motivation, and motor control in those with ADHD [2]. It affected 10.1% of school-age children and adolescents in Taiwan [3] as per ADHD’s diagnostic criteria in the Diagnostic and Statistical Manual of Mental Disorders, Fifth Edition (DSM-5) [4]. Children and adolescents with ADHD exhibit difficult behaviors, such as aggression, learning difficulties, rule breaking, low motivation, poor impulse control, and an inability for delayed gratification; such behavior increases their risks of depression, addiction, interpersonal difficulties, family disruption, academic and occupational underachievement, and suicide and risk-taking behavior [5]. However, ADHD is underdiagnosed and undertreated in Taiwan [6]. Research found that negative media coverage of pharmacological treatment partially accounts for undertreatment of ADHD in Taiwan [7]. Moreover, relative to males with ADHD, females with ADHD were more likely to be underdiagnosed and undertreated [8]. Further research is thus warranted for earlier diagnosis and improved treatment for children and adolescents with ADHD.

Research has discovered that many factors—including caregivers’ health beliefs, help-seeking behavior, and access to care—influence whether a child with emotional or behavioral problems receives treatment [9,10]. Caregivers’ beliefs regarding the causes of the child’s misbehavior will likely influence caregivers’ decisions about which type of interventions to pursue [11]. However, research on caregivers’ explanations for their child’s ADHD has yielded mixed results. According to a meta-analysis, some studies have noted that most caregivers consider ADHD to be a physical/medical illness; these caregivers attribute their child’s ADHD to biological causes, such as genetic causes and neurochemical imbalances [12]. Other studies have revealed that caregiver beliefs about the causes of ADHD diverge from the biomedical explanatory model, which is the scientific consensus. Reluctant to accept this model, these caregivers have attributed their child’s ADHD instead to difficulties with learning and memory or to indulgent caregiving from themselves or their spouse [12].

A study on caregivers of Iranian children with ADHD found that 52.3% of caregivers disagreed that ADHD is the result of biological and genetic reasons; 52.2% considered ADHD to be the result of an overindulgent caregiver [13]. A study evaluating caregiver-attributed etiology of ADHD in the United States revealed that 26% of caregivers believed that ADHD is the result of the overconsumption of sugar [14]. A study in the United Kingdom noted that caregivers attributed ADHD to a combination of biological and environmental causes, particularly genetic and dietary ones [15]. Research has suggested that caregiver attribution of ADHD to physical causes is positively associated with medication usage [16,17], whereas caregiver attribution of ADHD to sociological causes is negatively associated with medication usage [17,18]. Therefore, because caregivers decide treatments and service providers for their child with ADHD, it is important to understand such caregiver-attributed etiologies, particularly factors related to such attribution [19,20].

### 1.2. Factors Related to Caregiver Attributions of Children’s ADHD

Many factors have been noted to influence caregiver-attributed etiologies of their child’s ADHD. As for caregiver-related factors, mothers tended to attribute their child’s ADHD to biological reasons, whereas fathers tended to attribute ADHD to psychological factors, such as the child’s lack of effort [21]. Caregivers with a higher socioeconomic status were also more likely to attribute ADHD to genetic causes [11]. Culturally relevant factors, such as norms, medical approaches, beliefs, and values, influence the way members of various cultural groups view and respond to problematic behavior in children [22]. In some religion-related cultures, for example, the Muslim and Christian Lebanese cultures, hyperactivity and/or impulsivity in boys can be endorsed as typical by parents and viewed as gender preferred behavior [23].

As for child-related factors, the literature has furnished mixed results for the relationship between child’s sex and caregiver-attributed etiologies of ADHD. A study in Iran found no evidence for this relationship [24], whereas a U.S. study found that, relative to caregivers of girls, caregivers of boys were less likely to believe that stressful life events cause ADHD and more likely to believe in genetic causes [11]. Another study discovered that the onset of youth ADHD symptoms was associated with increased service use [25], but the relationship between youth ADHD symptoms and caregiver attributions of ADHD etiologies requires further study.

### 1.3. Role of Affiliate Stigma for Caregiver Attributions of Children’s ADHD

How affiliate stigma affects caregiver-attributed etiologies of ADHD remains unclear. Affiliate stigma, held by caregivers of people with mental illness, is defined as the perception, awareness, and internalization of public stigma toward those with mental illness [26]. Research has indicated that higher affiliate stigma in caregivers of children with ADHD is associated with poor caregiving outcomes [27] and that affiliate stigma in mothers of children with ADHD is positively related to the mother’s experience of distress [28]. Research has demonstrated that affiliate stigma has potential affective, cognitive, and behavioral consequences, such as feelings of subjective burden and stress in caregivers [26]. A review study noted that feelings of self-blame and being viewed as bad caregivers in the eyes of others were connected to the belief that ADHD is a result of poor caregiving [29]. Because both affiliate stigma and caregiver-attributed etiologies of ADHD potentially affect caregivers’ decisions on what help to seek for their child with ADHD, the relationship between these two factors requires further study.

### 1.4. Aims of the Present Study

The present study examined etiologies attributed by the caregivers of Taiwanese children with ADHD, particularly factors affecting such attribution. We had two hypotheses. First, caregivers attribute their child’s ADHD to various etiologies. Second, caregiver-attributed etiologies of ADHD are related to caregivers’ demographic characteristics and affiliate stigma and children’s demographic and ADHD characteristics.

## 2. Methods

### 2.1. Participants

Caregivers of children with ADHD were recruited between June 2018 and April 2019 from outpatient psychiatric clinics for children and adolescents of two medical centers in Kaohsiung, Taiwan. The child had to be aged 18 years or younger and had to have received a diagnosis of ADHD as per the DSM-5 criteria [4]. ADHD was diagnosed by two child psychiatrists through diagnostic interviews with children and caregivers. Multiple data sources were used to support each diagnosis, including clinical observations of each child’s behavior as well as caregiver’s ratings of ADHD symptoms on the short Chinese version of the Swanson, Nolan, and Pelham, Version IV Scale (SNAP-IV) [30,31]. Three child psychiatrists conducted a clinical interview with children and those who had an intellectual disability or autism spectrum disorder with difficulties in communication were excluded. Child psychiatrists also used a checklist to detect caregivers’ intellectual disability, schizophrenia, bipolar disorder, and any cognitive deficits that resulted in significant communication difficulties. Caregivers who had these conditions were excluded. This study invited 409 caregivers of children who received an ADHD diagnosis to participate. Of the 409, 9 (2.2%) declined to participate. Thus, 400 (97.8%) caregivers participated in the study. The Institutional Review Boards (IRBs) of Kaohsiung Medical University (KMUHIRB-E(I)-20180179) and Chang Gung Memorial Hospital, Kaohsiung Medical Center (201800723A3) approved this study.

### 2.2. Measures

#### 2.2.1. Caregiver-Attributed Etiologies of ADHD

First, information on common caregiver-attributed etiologies was gathered through three focus groups sessions; two groups for family caregivers and one group for child psychiatrists. The psychiatrists were interviewed on those caregiver-attributed etiologies they have encountered in their practice. Each group had five to eight members. On the basis of the information collected, we developed a self-report questionnaire. In the questionnaire, caregivers of children with ADHD rated how likely they perceived various etiologies of ADHD to currently be. The etiologies were brain dysfunction; failure of caregivers in disciplining the child; a poor diet, such as a sugar-rich diet; a poor living environment; the child imitating their peers’ misbehavior; failure of school staff in disciplining the child; the education system’s overemphasis on academic performance; and supernatural beings or divination-based reasons. Caregivers rated these etiologies on a four-point scale (1: *impossible*, 2: *unlikely*, 3: *somewhat likely*, and 4: *very likely*). Ratings of 3 and 4 were considered an attribution of ADHD to the etiology, and ratings of 1 and 2 were considered a nonattribution. Caregivers were further subdivided into those who attributed and did not attribute ADHD to the etiologies. We noted how many of these etiologies each caregiver attributed to their child’s ADHD.

#### 2.2.2. Affiliate Stigma Scale

The Affiliate Stigma Scale (ASS) is a self-rated 22-item questionnaire measuring the caregivers’ internalization of stigma toward their family member’s mental illness [26]. We focused on the caregiver’s affiliate stigma toward their child’s ADHD. The ASS included three domains: affect (comprising seven items, such as *I feel inferior because one of my children has ADHD*), cognition (comprising seven items, such as *My reputation is affected because I have a child with ADHD*), and behavior (comprising eight items, such as *I dare not tell others that I have a child with ADHD*). Respondents rated their current agreement with each statement in the ASS items on a four-point Likert scale (1: *strongly disagree*, 4: *strongly agree*). A higher score on the ASS indicates greater self-stigma toward their child’s ADHD. The original version was demonstrated to have excellent internal consistency (α = 0.94) and satisfactory predictive validity [26]. The ASS was also demonstrated to be a robust psychometric measure for Taiwanese populations [32]. In the present study, Cronbach’s α values for the ASS in general and its affect, cognition, and behavior domains were 0.95, 0.88, 0.89, and 0.89, respectively.

#### 2.2.3. Chinese Version of the SNAP-IV Scale, Parent Form

The short Chinese version of the SNAP-IV was used to assess the caregiver-reported severity of ADHD symptoms exhibited in the previous month. This version comprises 26 items encompassing the core DSM-derived ADHD subscales of inattention, hyperactivity, and impulsivity, and the oppositional symptoms of oppositional defiant disorder [30,31]. Each item is rated on a four-point Likert scale (0: *the symptom is absent*, 3: *the symptom is severe*). In the present study, Cronbach’s α values for inattention, hyperactivity, and impulsivity, and oppositional behavior were 0.89, 0.90, and 0.92, respectively.

#### 2.2.4. Factors Related to Caregivers and to Children

Information regarding caregivers’ and children’s sex, age, and education level was collected. This study also examined caregivers’ marital status (married, divorced, or separated). Caregivers’ occupational socioeconomic status (SES) was assessed using the Close-Ended Questionnaire of the Occupational Survey (CEQ-OS) [33], which classifies the occupational SES of the caregiver and their spouse into five levels, where a higher level indicates a higher occupational SES. The CEQ-OS has acceptable reliability and validity and has been used frequently in studies on children and adolescents in Taiwan [33]. Levels I, II, and III of the CEQ-OS were considered to indicate low occupational SESs, and levels IV and V were considered to indicate high occupational SESs [34]. Caregivers’ frequency of attending religious activities were classified into high (*frequently*) and low (*occasionally* or *never*).

### 2.3. Procedure and Statistical Analysis

Before starting the study, the principal investigator (PI) trained the research assistants to make sure that they were competent to direct the participants to complete the research questionnaire. Then, research assistants explained the procedures and methods of completing the questionnaire to the participants individually. The participants could propose any question when they had problems on completing the questionnaires, and the research assistants resolved their problems. The PI discussed with research assistants weekly to ensure the quality of the study.

SPSS 22.0 statistical software (SPSS Inc., Chicago, IL, USA) was used for data analysis. We calculated, for each etiology, the proportion of caregivers that attribute ADHD to it (number of caregivers attributing ADHD to the etiology/total number of caregivers). We also calculated the number of attributed etiologies for each caregiver. Forward conditional logistic regression was used to examine etiologies’ associations with caregivers’ and children’s sociodemographic characteristics, caregivers’ affiliate stigma, and children’s ADHD and oppositional symptoms. Significance was indicated using the odds ratio (OR) and 95% confidence interval (CI). Because multiple comparisons were conducted, a *p*-value <0.00625 (0.05/8) was considered to indicate significance. Multiple regression was used to analyze factors affecting the numbers of attributed etiologies for each caregiver. A two-tailed *p*-value <0.05 indicated statistical significance.

### 2.4. Ethics

The study procedures were conducted in accordance with the Declaration of Helsinki. The IRBs of Kaohsiung Medical University and Chang Gung Memorial Hospital, Kaohsiung Medical Center approved the study. All participants were informed about the study and provided written informed consent before completing research questionnaires.

## 3. Results

Table 1 presents the demographic characteristics, affiliate stigma, ADHD, and oppositional symptoms. Their scores of inattention and hyperactivity/impulsivity subscales on the SNAP-IV were 13.4 (3.6) and 9.8 (6.0), respectively, indicating mild levels of severities. Table 2 presents the proportion of etiologies that caregivers attributed to account for their child’s ADHD. The results indicated that the most commonly attributed etiology was brain dysfunction (84.8%), followed by failure of caregivers in disciplining the child (44.0%); a poor diet, such as a sugar-rich diet (40.8%); a poor living environment (38.8%); the child imitating their peers’ improper behavior (37.3%); failure of school staff in disciplining the child (29.0%); and the education system’s overemphasis on academic performance (27.3%). Only 3.8% of caregivers attributed their child’s ADHD to supernatural beings or divination-based reasons. There were 27 (6.8%), 115 (28.8%), and 258 (64.4%) participants who attributed their child’s ADHD to none, one, and two or more of the etiologies, respectively.

Table 3 presents the results for factors related to the extent of attribution of each etiology. Specifically, affiliate stigma was significantly and positively associated with the attributions of a poor diet, such as a sugar-rich diet (*p* < 0.001); a poor living environment (*p* = 0.005); the education system’s overemphasis on academic performance (*p* = 0.002); and supernatural beings or divination-based reasons (*p* = 0.003). Caregivers’ education duration was significantly and positively associated with the attribution of brain dysfunction (*p* = 0.002). Relative to caregivers of girls, caregivers of boys were more likely to attribute ADHD to brain dysfunction (*p* = 0.006). The severity of hyperactivity/impulsivity symptoms was positively and significantly associated with attributions of imitating peers’ misbehavior (*p* = 0.001). The severities of oppositional symptoms were significantly and positively associated with the attributions of caregivers’ (*p* = 0.001) and teachers’ failure to discipline the child (*p* = 0.001).

Table 4 presents results for factors related to the number of etiologies to which a caregiver attributes ADHD. Specifically, affiliate stigma (*p* = 0.001) and the severity of oppositional symptoms (*p* = 0.020) were both significantly and positively associated with the number of etiologies to which a caregiver attributes ADHD.

## 4. Discussion

This study found that caregivers attributed their child’s ADHD to a variety of etiologies. Caregivers’ education duration and children’s sex were related to the attribution of ADHD to brain dysfunction, whereas caregivers’ affiliate stigma and children’s oppositional and hyperactivity/impulsivity symptoms were related to the attribution of ADHD to nonbiological etiologies. Affiliate stigma and children’s oppositional symptoms were significantly associated with the number of etiologies to which a caregiver attributes their child’s ADHD.

### 4.1. Attribution of ADHD Etiologies

Because the caregivers of the present study were recruited from psychiatric outpatient clinics for children and adolescents, a high proportion (84.8%) of caregivers attributed ADHD to brain dysfunction. Notably, 64.4% of caregivers attributed their child’s ADHD to two or more etiologies, indicating that nonbiological etiologies are common explanations for ADHD. Research has demonstrated that caregiver attributions of ADHD etiologies predict their treatment choices [19]. The most common caregiver-attributed nonbiological etiology is the failure of caregivers in disciplining the child. This attribution potentially increases the caregiver’s feelings of self-blame and uncertainty [29], further exacerbating the care burden and psychological strain.

A high proportion of caregivers attributed their child’s ADHD to school-related etiologies, including failure of school staff in disciplining the child, the education system’s overemphasis on academic performance, and the child imitating their peers’ improper behavior. Being deeply influenced by Confucianism, Taiwanese people emphasize harmony, are oriented toward their community, and value academic achievement when raising their children [35]. However, the core symptoms of ADHD adversely affect a child’s ability to perform academically and to maintain harmonious relationships with others. Attributing a child’s ADHD to a failure of discipline by school staff may make cooperation between caregivers and school staff more difficult.

Similar to the results of U.S. studies [25,36], a high proportion of caregivers attribute ADHD to a poor diet, such as a sugar-rich diet. A recent systematic review and meta-analysis suggested that a diet that is high in refined sugar and saturated fat increases the risk of ADHD, whereas a healthy diet, characterized by a high consumption of fruits and vegetables, protects against ADHD or hyperactivity [37]. However, a recent birth cohort study noted no association between sucrose consumption and ADHD incidence between 6 and 11 years of age [38]. Furthermore, primarily attributing ADHD to diet may result in a low acceptance of medication and high acceptance of dietary interventions [36].

Only 3.8% of caregivers attributed their child’s ADHD to supernatural beings or divination-based reasons. Traditionally, spirituality and interdependence of human beings with the universe have always played a major role in influencing Chinese values and thoughts. Mental problems are at times likened to the evil spirits attaching to the individuals and requiring some form of penance and spiritual cleaning to regain mental wellbeing [39]. However, ADHD might be perceived as a behavioral and learning problem violating discipline and impeding academic performance, but not as a losing reality. Caregivers of children and adolescents with ADHD in this study were recruited from clinical units, and thus might have attributions of the etiologies of ADHD different from the traditional concepts.

### 4.2. Affiliate Stigma and Other Factors Related to the Attribution of Nonbiological Etiologies

The present study demonstrated that affiliate stigma is significantly associated with the attributions of several nonbiological etiologies of ADHD, including diet, a poor living environment, the education system’s overemphasis on academic achievement, and supernatural beings or divination-based reasons. Affiliate stigma was also significantly associated with the number of etiologies to which caregivers attributed their child’s ADHD. Affiliate stigma in caregivers of people with mental illness develops through the perception and internalization of public stigma toward caregivers [26]. Caregivers with intense affiliate stigma may feel negative emotions, such as shame and embarrassment, from internalized stigma [26,40]. A review study noted mixed results on the relationship between self-blame and the attribution of etiologies [29]. Our results suggest that the attribution of ADHD to diet, a poor living environment, educational stress, or supernatural beings reduces caregiver’s affiliate stigma and serves as a potential defense mechanism for their dignity. However, caregivers who attributed ADHD to nonbiological etiologies may delay medication treatment for their children with ADHD. This results in a vicious cycle where delayed treatment results in worsened symptoms, followed by worsened public prejudice, increased affiliate stigma, and finally more delayed treatment.

Our study demonstrated that the severities of hyperactivity/impulsivity symptoms are significantly and positively associated with the caregiver attributing ADHD to the child imitating their peers’ misbehavior. According to the common-sense model of illness danger [41], when one encounters a health threat, the identification of an illness’ symptoms and reasons are the first and second processes of cognitive representation, respectively, that guide one’s choice of coping strategies. Because children commonly exhibit hyperactivity and impulsivity, caregivers may attribute their child’s ADHD to the child’s imitation of their peers’ misbehaviors. We also found that the severity of oppositional symptoms was significantly and positively associated with the attributions of ADHD to failure of discipline on the part of caregivers and teachers. Opposition to authority is a common comorbid presentation in ADHD, where the display of defiant, vindictive, angry, and argumentative behavior toward authority figures can be considered a conscious attempt at rebelling against the caregiver’s or teacher’s discipline. Oppositional symptoms may also complicate the presentations of ADHD and thus confuse caregivers’ judgement on reasons behind ADHD’s development.

### 4.3. Factors Related to the Attribution of Brain Dysfunction

Caregivers of boys or caregivers who were educated for longer were significantly more likely to attribute ADHD to brain dysfunction. A U.S. study found that parents of boys were more likely than parents of girls to endorse genetic causes and less likely to cite stressful life events as ADHD causes [11]. An interpretation of ADHD symptoms as a temporary adjustment phenomenon may contribute to parents’ determination that no professional interventions are required [42]. Further investigation is required into the effect of gendered social constructs on the difference in attributions of ADHD etiologies and further intervention between caregivers of boys and caregivers of girls. Generally, caregivers who are educated for longer tend to be better able to understand and access authoritative information on ADHD’s etiology. Previous studies have found that people with higher levels of education were more likely to have heard of ADHD [43] and identify ADHD correctly [44]. The results of the present study further supported the role of educational duration for attribution of brain dysfunction. Furthermore, we found that occupational SES did not relate to attribution of ADHD etiologies. Caregivers who had a high occupational SES had longer education duration than those with low occupational SES in the present study (15.3 years vs. 12.9 years, *t* = 8.783, *p* < 0.001); therefore, the association of occupational SES with attribution of etiologies might be decreased by education duration in forward conditional logistic regression.

## 5. Limitations

Our study has several limitations. First, our cross-sectional research design limited the inference of a causal relationship between affiliate stigma and the attribution of ADHD to various etiologies. Second, our data were obtained solely from caregiver self-reports, potentially resulting in common-method variance. Third, we did not examine the effects of multidimensional psychosocial factors, such as social support or health beliefs, on the attribution of ADHD to various etiologies. Fourth, this cross-sectional study recruited caregivers of children with ADHD diagnosed for various time periods, which might influence caregivers’ attribution of ADHD etiologies. Fifth, the present study did not examine the effect of residential areas of families on their attributions. However, research has demonstrated that the Internet becomes the first and most popular source for caregivers to search for opinions regarding the etiologies and intervention models of ADHD [45]. The internet may reduce the limitation of assessing ADHD-related knowledge for those living in rural areas.

## 6. Conclusions

The debate on ADHD’s causes centers on the opposition between neurobiological and psychosocial explanations [46]. We demonstrated that caregivers of children with ADHD attributed their child’s ADHD to a variety of etiologies. Because caregivers and health professionals may have divergent beliefs when working together to diagnose and treat children with ADHD [47], their shared decision-making process can be enhanced by a better understanding of caregiver-attributed etiologies of ADHD. Caregivers’ affiliate stigma and education levels and children’s sex, hyperactivity/impulsivity, and oppositional symptoms were related to various etiologies’ extent of attribution. Health professionals should account for these factors when working with caregivers to develop treatment plans for children with ADHD.

## Figures and Tables

**Table 1 ijerph-17-01652-t001:** Demographic characteristics, affiliate stigma, ADHD, and oppositional symptoms (N = 400).

	*n* (%)	Mean (SD)	Range
Caregivers			
Relationship with the child			
Mother	287 (71.8)		
Father	90 (22.5)		
Others	23 (5.8)		
Age (years)		43.4 (6.8)	25–70
Sex			
Female	304 (76.0)		
Male	96 (24.0)		
Education duration (years)		13.8 (2.9)	3–23
Caregiver marriage status			
Married	320 (80)		
Divorced or separated	80 (20)		
Occupational socioeconomic status			
High	155 (38.8)		
Low	245 (61.2)		
Frequency of attending religious activities			
High	143 (35.8)		
Low	257 (64.3)		
Affiliate stigma on the ASS		38.7 (11.3)	22–75
Children			
Age (years)		10.7 (3.2)	4–18
Sex			
Girls	64 (16.0)		
Boys	336 (84.0)		
ADHD and oppositional symptoms on the SNAP-IV			
Inattention		13.4 (3.6)	0–27
Hyperactivity/impulsivity		9.8 (6.0)	0–27
Opposition		10.1 (6.0)	0–24

ADHD: attention-deficit/hyperactivity disorder; ASS: Affiliate Stigma Scale; SNAP-IV: Swanson, Nolan, and Pelham, Version IV Scale.

**Table 2 ijerph-17-01652-t002:** Caregiver-attributed etiologies of ADHD (N = 400).

	*n* (%)
Brain dysfunction	339 (84.8)
Failure of caregivers in disciplining the child	176 (44.0)
Poor diet, such as a sugar-rich diet	163 (40.8)
Poor living environment	155 (38.8)
Child’s imitating peers’ misbehavior	149 (37.3)
Failure of school staff in disciplining the child	116 (29.0)
The education system’s overemphasis on academic achievement	109 (27.3)
Supernatural beings or divination-based reasons	15 (3.8)
Number of ADHD etiologies attributed by caregivers	
0	27 (6.8)
1	115 (28.8)
2	74 (18.5)
3	53 (13.3)
4	58 (14.5)
5	39 (9.8)
6	28 (7.0)

**Table 3 ijerph-17-01652-t003:** Factors related to the etiologies of ADHD attributed by caregivers.

	Brain Dysfunction		Caregivers’ Failure to Discipline		Poor Diet, such as a Sugar-Rich Diet		Poor Living Environment		Imitation of Peers’ Misbehavior		Teachers’ Failure to Discipline		Overemphasis on Academic Achievement		Supernatural Beings or Divination-Based Reasons	
	OR (95% CI)	*p*	OR (95% CI)	*p*	OR (95% CI)	*p*	OR (95% CI)	*p*	OR (95% CI)	*p*	OR (95% CI)	*p*	OR (95% CI)	*p*	OR (95% CI)	*p*
*Caregivers*																
Male ^a^			0.518 (0.314–0.855)	0.010	0.598 (0.360–0.993)	0.047	0.533 (0.320–0.888)	0.016								
Age									0.963 (0.932–0.994)	0.021						
Education duration	1.168 (1.059–1.287)	**0.002**														
Low frequency of attending religious activities ^b^															0.300 (0.102–0.885)	0.029
Affiliate stigma			1.020 (1.001–1.039)	0.039	1.994 (1.436–2.769)	**<0.001**	1.027 (1.008–1.046)	**0.005**			1.025 (1.004–1.045)	0.017	1.032 (1.011–1.052)	**0.002**	1.071 (1.023–1.121)	**0.003**
*Children*																
Male ^c^	2.372 (1.238–4.545)	**0.006**														
Hyperactivity/impulsivity									1.060 (1.023–1.098)	**0.001**	0.954 (0.912–0.999)	0.044				
Opposition			1.060 (1.023–1.098)	**0.001**							1.079 (1.032–1.128)	**0.001**				

^a^: female as reference; ^b^: high frequency of attending religious activities as reference; ^c^: female as reference.

**Table 4 ijerph-17-01652-t004:** Factors related to the numbers of etiologies of ADHD attributed by the caregivers.

	Beta	*t*	*p*
Caregivers’ sex: males ^a^	−0.076	−1.497	0.135
Caregivers’ age	−0.020	−0.373	0.709
Caregivers’ education duration	0.030	0.552	0.581
Caregivers’ marriage: divorced or separated ^b^	−0.090	−1.818	0.070
Low frequency of caregivers’ attending religious activities ^c^	−0.069	−1.383	0.168
Low caregivers’ occupational socioeconomic status ^d^	0.050	0.918	0.359
Caregivers’ affiliate stigma	**0.168**	**3.224**	**0.001**
Children’s sex: boys ^e^	0.056	1.124	0.262
Children’s age	0.024	0.442	0.659
Children’s inattention symptoms	−0.029	−0.474	0.636
Children’s hyperactivity/impulsivity symptoms	0.025	0.365	0.716
Children’s opposition symptoms	**0.150**	**2.339**	**0.020**

^a^: female as reference; ^b^: married as reference; ^c^: high frequency of attending religious activities as reference; ^d^: high socioeconomic status as reference; ^e^: girls as reference.

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
