# Peer review of "Caregiver-Attributed Etiologies of Children’s Attention-Deficit/Hyperactivity Disorder: A Study in Taiwan"

_ijerph, 2020, doi:10.3390/ijerph17051652_

Round 1

Reviewer 1 Report

I appreciate the opportunity to review the manuscript entitled "Caregiver-Attributed Etiologies of Children's Attention-Deficit/Hyperactivity Disorder: A Study in Taiwan". The study examines the caregivers of children with ADHD attribution about the etiology of ADHD and the factors that might have impact on this attribution. It is an interesting topic, since the results might affect several conclusions, for example the treatment of ADHD - as it has been mentioned by the authors also. The manuscript is well structured and is easily readable. However there are some parts which do not contain information what would be essential.

Abstract
The applied measures should also be mentioned in the Abstract.

Introduction

In the beginning of this chapter the authors mention that ADHD is a neurodevelopmental disorder, however the scientifically proven etiology of ADHD is lacking.

The authors refer that ADHD is underdiagnosed and undertreated in Taiwan. Are there some researches about the reasons of these facts?

Procedure and participants

Concerning the procedure itself the description is moderately deficient. More information are needed regarding the exact process.

By what time did the information gathering about the attribution and related parameters started? Right after the child got the diagnose of ADHD or when and how? Please include all these information.

What was the distribution of the types of ADHD by the children? Was the type of ADHD taken into account in the regression analysis?

As for the exclusions, how was the child’s intellectual disability and autism spectrum disorder with difficulties in communication measured? Also by the diagnostic interview? How was the intellectual disability, schizophrenia, bipolar disorder, or any cognitive deficits that resulted in significant communication difficulties measured in the caregivers?

Did the authors measure other psychiatric diagnoses that are commonly comorbid with ADHD (for example conduct disorder, oppositional defiant disorder, etc.)?

Measures

Regarding the Affiliate Stigma Scale (in line 123), in the first sentence, reference is missing.

The SNAP-IV abbreviation is not introduced.

What are the justifications of measuring the attending religious activities? No research was mentioned in the Introduction regarding this measure. Moreover, how might it affect the attribution of the caregivers for ADHD?

Results

In the first sentence (line 166 and 167) there are too much „and”, it should be rewritten, while the sentence is too long. This is also the case about the name of Table 1.

Where significant results were found the exact statistical parameters should be reported as well.

Education duration should be explained in the text while "Years" are represented in the table. Different terms are misleading.

In the „4.3. Factors related to the attribution of brain dysfunction” section references are needed to connect the results with the existing literature.

Limitations

Limitations should be numbered as 5. 

Conclusion

Conclusion should be numbered as 6.

Author Response

We appreciate your comments on our manuscript. As discussed below, we have revised our manuscript with underlines according to the reviewers. The following responses have been prepared to address your comments in a point-by-point fashion. Please let us know if there is anything else we should provide.

Comment

The applied measures should also be mentioned in the Abstract.

Response

Thank you for your reminding. We added them into Abstract section as below. Please refer to line 22-23.

“Caregivers completed the self-report questionnaire to rate how likely they perceived various etiologies of ADHD to be, the Affiliate Stigma Scale for the level of affiliate stigma, and the short Chinese version of the Swanson, Nolan, and Pelham, Version IV Scale for child’s ADHD and oppositional symptoms.”

Comment

In the beginning of this chapter the authors mention that ADHD is a neurodevelopmental disorder, however the scientifically proven etiology of ADHD is lacking.

Response

We added the results of previous studies on scientifically proven etiology of ADHD into the revised manuscript as below. Please refer to line 39-43.

“Attention-deficit/hyperactivity disorder (ADHD) is the most prevalent childhood-onset neurodevelopmental disorder. ADHD is highly heritable and multifactorial; multiple genes and non-inherited factors contribute to the disorder [1]. Neuropsychological dysfunctions such as executive dysfunction have been proposed to explain the deficits in behavioral inhibition, working memory, regulation of motivation, and motor control in those with ADHD [2].”

Comment

The authors refer that ADHD is underdiagnosed and undertreated in Taiwan. Are there some researches about the reasons of these facts?

Response

We added the results of previous studies in Taiwan for the possible reasons as below. Please refer to line 48-51.

“Research found that negative media coverage of pharmacological treatment partially account for undertreatment of ADHD in Taiwan [7]. Moreover, relative to males with ADHD, females with ADHD were more likely to be underdiagnosed and undertreated [8].”

Comment

Concerning the procedure itself the description is moderately deficient. More information are needed regarding the exact process.

Response

We added more information regarding the procedures of this study into the revised manuscript as below. Please refer to line 163-167.

“Before starting the study, the PI trained the research assistants to make sure that they were competent to direct the participants to complete the research questionnaire. Then research assistants explained the procedures and methods of completing the questionnaire to the participants individually. The participants could propose any question when they had problems on completing the questionnaires, and the research assistants resolved their problems. The PI discussed with research assistants weekly to make sure the quality of the study.”

Comment

By what time did the information gathering about the attribution and related parameters started? Right after the child got the diagnose of ADHD or when and how? Please include all these information.

Response

  1. The present study evaluated caregivers’ current attributions of etiologies of ADHD and affiliate stigma, as well as child’s ADHD symptoms in the previous month. We emphasized them in the revised manuscript. Please refer to line 126, 140 and 148.
  2. This cross-sectional study recruited caregivers of children with ADHD diagnosed for various time periods, which may influence caregivers’ attribution of ADHD etiologies. Because of time lag between the first time of diagnosing ADHD and this survey, we added it as one of limitations of this study. Please refer to line 318-320.

Comment

What was the distribution of the types of ADHD by the children? Was the type of ADHD taken into account in the regression analysis?

Response

The present study recruited the caregivers of children with ADHD diagnosed based on DSM-5 criteria. The DSM-5 has removed the subtypes for ADHD. Therefore, we used the SNAP-IV to measure the severities of inattention and hyperactivity/impulsivity. Because of children recruited from the child and adolescent psychiatric clinics and receiving treatment, their scores of inattention and hyperactivity/impulsivity subscales on the SNAP-IV were 13.4 (3.6) and 9.8 (6.0), respectively, indicating mild levels of severities. Both inattention and hyperactivity/impulsivity symptoms were selected into regression analysis. We added the introduction in the revised manuscript. Please refer to line 183-185.

Comment

As for the exclusions, how was the child’s intellectual disability and autism spectrum disorder with difficulties in communication measured? Also by the diagnostic interview? How was the intellectual disability, schizophrenia, bipolar disorder, or any cognitive deficits that resulted in significant communication difficulties measured in the caregivers?

Response

Thank you for your reminding. We added explanations for the methods to detect caregivers’ and children’s psychiatric morbidities in the revised manuscript as below. Please refer to line 112-116.

“Three child psychiatrists conducted a clinical interview with children and those who had an intellectual disability or autism spectrum disorder with difficulties in communication were excluded. Child psychiatrists also used a checklist to detect caregivers’ intellectual disability, schizophrenia, bipolar disorder, and any cognitive deficits that resulted in significant communication difficulties. Caregivers who had these conditions were excluded.”

Comment

Did the authors measure other psychiatric diagnoses that are commonly comorbid with ADHD (for example conduct disorder, oppositional defiant disorder, etc.)?

Response

We did not measure other psychiatric diagnoses that are commonly comorbid with ADHD. However, we used the SNAP-IV to measure children’s opposition symptoms. The results found that the severities of oppositional symptoms were significantly and positively associated with the attributions of caregivers’ and teachers’ failure to discipline the child, as well as that the severity of oppositional symptoms were both significantly and positively associated with the number of etiologies a caregiver attributes ADHD to.

Comment

Regarding the Affiliate Stigma Scale (in line 123), in the first sentence, reference is missing.

Response

Thank you for your reminding. We added the reference into the revised manuscript. Please refer to line 136.

Comment

The SNAP-IV abbreviation is not introduced.

Response

We introduced the SNAP-IV abbreviation in Introduction section. Please refer to line 111-112.

Comment

What are the justifications of measuring the attending religious activities? No research was mentioned in the Introduction regarding this measure. Moreover, how might it affect the attribution of the caregivers for ADHD?

Response

Thank you for your reminding. We added a paragraph to introduce why we examined the role of religion beliefs for attribution of ADHD etiologies as below. Please refer to line 77-81.

“Culturally relevant factors, such as norms, medical approaches, beliefs, and values influence the way members of various cultural groups view and respond to problematic behavior in children [22]. In some religion-related cultures, for example, the Muslim and Christian Lebanese, hyperactivity and/or impulsivity in boys can be endorsed as typical by parents and viewed as gender preferred behavior [23].”

Comment

In the first sentence (line 166 and 167) there are too much „and”, it should be rewritten, while the sentence is too long. This is also the case about the name of Table 1.

Response

We shortened the sentence and the names of Table 1 as below. Please refer to line 183 and line 194.

“Table 1 presents demographic characteristics, affiliate stigma, ADHD and oppositional symptoms.”

Comment

Where significant results were found the exact statistical parameters should be reported as well.

Response

We added p values into the text of Results section to show the significant results. Please refer to line 200-208 and 213-214.

Comment

Education duration should be explained in the text while "Years" are represented in the table. Different terms are misleading.

Response

We changed “education level” into “education duration” in the text (line 203), Table 1, Table 3 and Table 4.

Comment

In the „4.3. Factors related to the attribution of brain dysfunction” section references are needed to connect the results with the existing literature.

Response

We added the references and discussion on the effects of caregivers’ education and children’s sex into section 4.3. as below. Please refer to line 296-302 and 304-307

“A US study found that parents of boys were more likely than parents of girls to endorse genetic causes and less likely to cite stressful life events as ADHD causes [11]. An interpretation of ADHD symptoms as a temporary adjustment phenomenon may contribute to parents’ determination that no professional interventions are required [42]. Further investigation is required into the effect of gendered social constructs on the difference in attributions of ADHD etiologies and further intervention between caregivers of boys and caregivers of girls.”

“Previous studies have found that people with higher levels of education were more likely to have heard of ADHD [43] and identify ADHD correctly [44]. The results of the present study further supported the role of educational duration for attribution of brain dysfunction. Furthermore, we found that occupational SES did not relate to attribution of ADHD etiologies. Caregivers who had a high occupational SES had longer education duration than those with low occupational SES in the present study (15.3 years vs. 12.9 years, t = 8.783, p < 0.001); therefore, the association of occupational SES with attribution of etiologies might be decreased by education duration in forward conditional logistic regression.”

Comment

Limitations should be numbered as 5. 

Response

We numbered “Limitations” as 5. Please refer to line 312.

Comment

Conclusion should be numbered as 6.

Response

We numbered “Conclusion” as 6. Please refer to line 325.

Reviewer 2 Report

An important topic, especially following the release of the new AAP and SDBP Guidelines for Complex ADHD, which underscore the importance of psychosocial treatment as foundational in ADHD management.  It will be critical to understand parents' reservations about treatment and help them through them to give their children appropriate care.  I was particularly intrigued by the discussion of subtypes in family attribution style suggested by table 3.  I have just a few questions about this very well-done survey study and look forward to seeing its needed findings in print:

  1. I was surprised that SES was not a more significant driver of findings.  Did the authors do a zero order correlation to check for multicollinearity among variables?  I am also curious about the breakdown of participating families from urban vs. rural areas and whether that had some effect on their attributions.

  2. A bit more explanation is needed about what is meant by spirituality and divination, especially in comparison to the word count allocated to other attributions discussed in this paper.  I can surmise that it reflects old traditions and customs, but I do not know if it will necessarily be self-evident to readers.  

Author Response

We appreciate your comments on our manuscript. As discussed below, we have revised our manuscript with underlines according to the reviewers. The following responses have been prepared to address your comments in a point-by-point fashion. Please let us know if there is anything else we should provide.

Comment

I was surprised that SES was not a more significant driver of findings.  Did the authors do a zero order correlation to check for multicollinearity among variables?  I am also curious about the breakdown of participating families from urban vs. rural areas and whether that had some effect on their attributions.
Response

  1. Thank you for your reminding. We found a significant correlation between caregivers’ occupational SES and education duration (Spearman’s correlation = 0.422, p <0.001). Caregivers who had a high occupational SES had longer education duration than those with low occupational SES in the present study (15.3 years vs. 12.9 years, t = 8.783, p < 0.001). Therefore, the present study used forward conditional logistic regression analysis to reduce the influence of collinearity between occupational SES and education duration on attribution of etiologies. We added explanation as below. Please refer to line 307-311.

“Furthermore, we found that occupational SES did not relate to attribution of ADHD etiologies. Caregivers who had a high occupational SES had longer education duration than those with low occupational SES in the present study (15.3 years vs. 12.9 years, t = 8.783, p < 0.001); therefore, the association of occupational SES with attribution of etiologies might be decreased by education duration in forward conditional logistic regression.”

  1. The present study did not examine the effect of participating families from urban vs. rural areas on their attributions. We listed it as one of limitations of this study as below. We also emphasize the role of the internet for assessing ADHD-related knowledge among caregivers living in rural areas. Please refer to line 320-324.

“Fifth, the present study did not examine the effect of residential areas of families on their attributions. However, research has demonstrated that the internet becomes the first and most popular source for caregivers to search for opinions regarding the etiologies and intervention models of ADHD [45]. The internet may reduce the limitation of assessing ADHD-related knowledge for those living in rural areas.”

Comment

A bit more explanation is needed about what is meant by spirituality and divination, especially in comparison to the word count allocated to other attributions discussed in this paper.  I can surmise that it reflects old traditions and customs, but I do not know if it will necessarily be self-evident to readers.  

Response

We added a new paragraph to discuss the low rate of attributing ADHD to spirituality and divination found in the present study as below. Please refer to line 253-261.

“Only 3.8% of caregivers attributed their child’s ADHD to supernatural beings or divination-based reasons. Traditionally, spirituality and interdependence of human beings with the universe have always played a major role in influencing Chinese values and thoughts. Mental problems are at times likened to the evil spirits attaching to the individuals and requiring some form of penance and spiritual cleaning to regain mental wellbeing [39]. However, ADHD might be perceived as behavioral and learning problem violating discipline and impeding academic performance but not as losing reality. Caregivers of children and adolescents with ADHD in this study were recruited from clinical units and thus might have attributions of the etiologies of ADHD different from the traditional concepts.”